# Positive epistasis drives clavulanic acid resistance in double mutant libraries of BlaC β-lactamase
Marko Radojković[1] & Marcellus Ubbink [1] ✉

Phenotypic effects of mutations are highly dependent on the genetic backgrounds in which they occur, due to epistatic effects. To test how easily the loss of enzyme activity can be compensated for, we screen mutant libraries of BlaC, a β-lactamase from *Mycobacterium tuberculosis*, for fitness in the presence of carbenicillin and the inhibitor clavulanic acid. Using a semi-rational approach and deep sequencing, we prepare four double-site saturation libraries and determine the relative fitness effect for 1534/1540 (99.6%) of the unique library members at two temperatures. Each library comprises variants of a residue known to be relevant for clavulanic acid resistance as well as residue 105, which regulates access to the active site. Variants with greatly improved fitness were identified within each library, demonstrating that compensatory mutations for loss of activity can be readily found. In most cases, the fittest variants are a result of positive epistasis, indicating strong synergistic effects between the chosen residue pairs. Our study sheds light on a role of epistasis in the evolution of functional residues and underlines the highly adaptive potential of BlaC.

Throughout the course of evolution, amino acid sequences of enzymes had to be tuned to adapt to ever-changing environmental conditions. Substitutions providing advantageous phenotypes are more likely to be fixed and those which are compromising biological function are less likely to be retained. Thus, apart from functional traits such as substrate affinity, catalytic efficiency and allosteric modulation, enzymes have a property called evolvability; the ability to evolve to a new or improved function. Because selective pressures fluctuate over time, evolution will favor enzymes that retain a degree of evolvability and buffering capacity to accommodate new mutations. Predicting evolutionary trajectories from the effects of individual mutations is notoriously difficult due to epistasis, a phenomenon which occurs when two or more mutations interact in a nonadditive manner, resulting in an unpredicted phenotype that can compensate for deleterious mutations or abrogate beneficial ones[1–3]. Intragenic epistasis increases the complexity of mutational landscapes[4–8]. The magnitude of epistasis and its implication in the evolution of enzymes are still being unraveled, often with contrary views[9–15].

β-Lactamases are a group of enzymes that can hydrolyze β-lactam antibiotics, and display a high degree of evolvability, which makes them powerful systems for evolution studies. Here, we focus on BlaC, the β-lactamase from *Mycobacterium tuberculosis* (Mtb). BlaC is a broad-spectrum Ambler class A β-lactamase encoded by a chromosomal gene[16]. Evolvability of BlaC is constrained by numerous factors, such as efficient folding and export, stability at 37 °C, and the ability to hydrolyze a wide spectrum of β-lactam antibiotics. β-Lactam/β-lactamase inhibitor combinations are considered for the treatment of tuberculosis[17–20], but the degree to which BlaC is able to circumvent inhibition is still unknown and will require strict monitoring. One of the most commonly used β-lactam inhibitors in clinical practice is clavulanic acid, which forms a covalent intermediate with Ser70 (standard Ambler numbering[21]). Secondary chemistry in clavulanic acid results in either very slow hydrolysis or the formation of a non-hydrolysable adduct[22,23]. Several studies demonstrated that clavulanic acid inhibition can be strongly reduced in BlaC by introducing single amino acid substitutions[24–26]. The effect was even more pronounced for double mutants[24,25]. Though these studies showed reduced sensitivity for clavulanic acid in vitro, the introduction of the *blaC* gene with the single or double mutations into a Δ*blaC* Mtb strain did not result in any significant in vivo advantage over the wild-type, questioning the relevance of these mutations in a clinical setting[24]. The likely reason is that most of the inhibitor-resistant mutations also severely cripple catalytic efficiency, with K234R being an exception[25]. Mutations that reduce sensitivity for clavulanic acid, therefore, require compensatory mutations to enhance catalytic activity. One mutation that was found to ameliorate activity of BlaC is I105F[27]. Position 105 has been dubbed the gatekeeper residue[27], as it regulates access to the enzymes active-site, and is in most class A β-lactamases occupied by an amino acid with an aromatic side chain. The I105F variant showed both higher in vitro activity and in vivo resistance against ampicillin than wild-type BlaC[27]. We

[1]Leiden Institute of Chemistry, Leiden University, Einsteinweg 55, 2333 CC Leiden, The Netherlands. ✉e-mail: m.ubbink@chem.leidenuniv.nl

wondered whether this mutation could act synergistically and compensate for the loss of activity caused by inhibitor-resistant mutations, potentially yielding variants with greatly improved in vivo resistance.

For that reason, we coupled gatekeeper residue (Ile105) with positions which have been linked to reduced sensitivity for clavulanic acid (Ser130, Gly132, Arg220 or Lys234)[24–26] and investigated their potential to confer clavulanate resistance in a cellular assay. To get a complete topological map of mutational landscapes, we sampled all possible amino acid combinations between these residue pairs and prepared four double-site saturation libraries. Furthermore, we wanted to test whether temperature can affect evolutionary outcomes, because the evolution of new function is often accompanied by a reduction of stability, so evolvability is expected to be larger at a lower temperature[28–31]. Using deep sequencing and *Escherichia coli* fitness as a proxy for Mtb fitness, relative fitness effects for 1534/1540 (99.6%) unique library variants were determined at 37 °C and 30 °C. The data show that many mutations of the gatekeeper residue improve fitness, both individually and in combination with other positions, whose mutational tolerance shows good correlation with their conservation profile. Mutational landscapes at 30 °C and 37 °C were highly similar, suggesting that temperature does not significantly impact evolutionary outcome. The majority of the fittest variants exhibited positive epistasis, which was 4.4 times less pervasive than negative epistasis. Findings presented in this study reveal the formidable compensatory potential of the gatekeeper residue and emphasize its role in the creation of highly resistant bacterial phenotypes. The roles of the individual residues for BlaC are discussed.

## Results
### Preparation of deep mutant libraries and clavulanic acid selection
We used the nicking mutagenesis approach[32] to create four double-site saturation libraries of the gatekeeper residue (Ile105) in combination with different inhibitor-resistant hotspots (Ser130, Gly132, Arg220 and Lys234). Two positions in each library were mutated to all other 19 amino acids effectively yielding libraries comprising 400 variants, with 20 variants being present in every library (variation of position 105 only), which yields 1540 unique combinations of amino acids.

To evaluate the fitness of all variants under the combined selection pressures of the β-lactam antibiotic carbenicillin and the β-lactamase inhibitor clavulanic acid, libraries were introduced into *E. coli* cells. Bacterial fitness in the presence of β-lactam antibiotic, i.e. the ability of bacteria to survive and reproduce, predominantly lies on beta-lactamase activity[33–36]. Here, we correlate bacterial fitness with enzyme activity and its ability to evade inhibition. Inoculated liquid cultures without (no selection) or with carbenicillin and clavulanic acid at the minimal inhibitory concentration (MIC) of wild-type (100 µg/mL carbenicillin and 1 µg/mL clavulanic acid and) were incubated at 37 °C or 30 °C. Slower growth at 30 °C was compensated for by a longer incubation time.

Deep sequencing was performed using Illumina 150 bp or 300 bp paired-end sequencing (PE) and counts for each variant were obtained for conditions with and without selection. After quality filtering, an average of approximately 2200- and 760-fold depth of coverage per sequenced library sample was obtained for 150 PE (libraries 1 and 2) and 300 PE (libraries 3 and 4) reads, respectively. The relative fitness effect ($F_i$) of each variant was determined as previously described in work of Stiffler et al.[35], where $F_i$ is calculated as the logarithm of the ratio of the allele counts in the selected ($N_i^{sel}$) and unselected populations ($N_i^{unsel}$), relative to the wild-type allele, as shown in Eq. 1:

$$F_i = \log_{10}\left[\frac{N_i^{sel}}{N_i^{unsel}}\right] - \log_{10}\left[\frac{N_i^{wt,sel}}{N_i^{wt,unsel}}\right] \qquad (1)$$

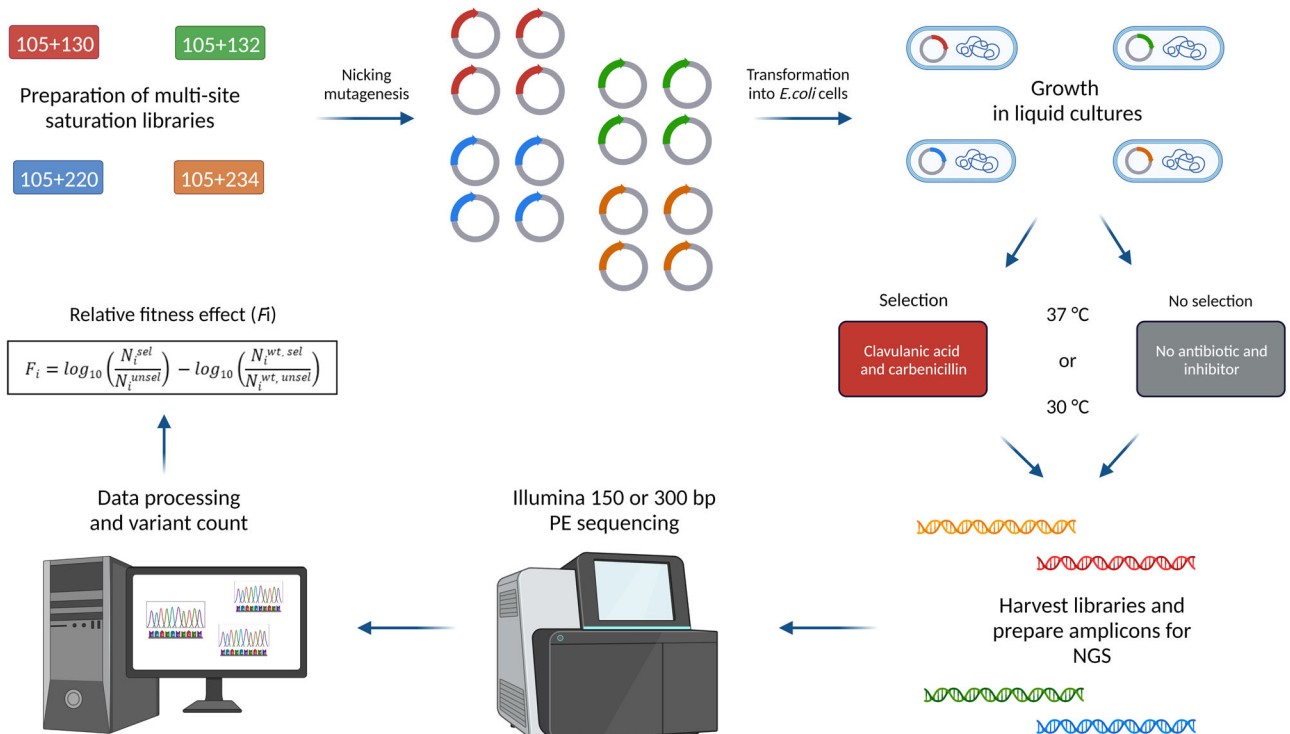

**Fig. 1 | Schematic illustration of the workflow.** Plasmids carrying wild-type *blaC* gene were mutated by saturating the selected positions with all possible amino acid replacements using nicking mutagenesis. The resulting double-site saturation libraries were introduced into *E. coli* cells for growth in selected (clavulanic acid and carbenicillin) and unselected conditions, at 37 °C or 30 °C. Harvested plasmid libraries were used as templates to prepare amplicons for deep sequencing on Illumina 150 bp or 300 bp PE platforms. Data were processed and the relative fitness effect of each variant was calculated as the ratio of reads obtained from selected libraries ($N_i^{sel}$) versus unselected libraries ($N_i^{unsel}$), relative to the wild-type values. Created with Biorender.com.

Variants that show increased or decreased fitness relative to the wild-type have positive or negative $F_i$ values, respectively. $F_i$ values close to the wild-type ($F_i \approx 0$) represent variants with a neutral fitness effect. A good correlation of $F_i$ values was observed between two independent datasets ($r_{avg} = 0.9$; Fig. S1). An overview of the workflow is shown in Fig. 1.

## Fitness landscape at 37 °C

Figure 2 shows the relative fitness effect of single mutant variants from all four libraries selected at 37 °C. Surprisingly, a myriad of Ile105 substitutions result in a beneficial phenotype. When averaged over four libraries, 13/19 (68%) variants are significantly better than the wild-type, with only one variant being deleterious. Unexpectedly, Arg105 was found to be the best option at the gatekeeper position among all single mutant variants (Fig. 2a). On the contrary, replacement of the wild-type residue at positions which are known to reduce inhibitor sensitivity mostly result in a deleterious phenotype. However, the mutational tolerance of these positions differs substantially (Fig. 2b–e). Arginine at position 220 seems to be most tolerant to substitutions, where 9/19 (47%) mutations are neutral, whereas Ser130 and Lys234 are the positions with the least number of viable substitutions, with the former having only one neutral (S130G), and the latter only one beneficial mutation (K234R). The only other variant with a beneficial phenotype was found for position 132 (G132A), together with two variants with a neutral fitness effect (Fig. 2c).

Figure 3 displays the complete fitness landscape for all four libraries. It is immediately apparent that most of the double mutations in each library have a detrimental effect on fitness, with library 3 being an exception (Fig. 3c). In most cases, the strongest phenotypes are observed when aromatic amino acids, histidine or arginine occupy position 105, which are in some cases even compensating for the deleterious effect of the other mutation (Fig. 3c). Surprisingly, Gly105 in combination with Asn132 is the

fittest variant in library 2 (Fig. 3b). Variants with a beneficial phenotype are most abundant in library 3 (Fig. 3c), whereas libraries 1 and 4 have the least number of variants with a positive effect on fitness (Figs. 3a and d, respectively), which is consistent with the number of viable substitutions observed for single mutant variants (Fig. 2).

## Comparison of selection at different temperatures

To inspect what role temperature plays in the shaping of fitness landscapes, we also determined the fitness effect of libraries selected at 30 °C and compared these to the data obtained at 37 °C (Fig. 4). Increased fitness values were observed for all four libraries selected at 30 °C, although to a different extent. The effect is most pronounced for library 3 (Fig. 4c), whilst moderately improved fitness is observed for libraries 1 and 4 (Figs. 4a and d). The lower temperature had the least overall positive impact on library 2 (Fig. 4b). The global increase in fitness values at 30 °C, as compared to 37 °C, is clearly borne out when plotting a histogram of the distribution of fitness effects for each library (Fig. 5). Libraries 1, 3 and 4 show a clear positive shift in fitness values, whereas library 2 exhibits only a small shift.

The increase in fitness values of variants at 30 °C could be a consequence of two factors. First, the balance between activity and stability is more optimal at 30 °C. Second, wild-type BlaC is performing worse at 30 °C than at 37 °C, i.e., the combination of carbenicillin and clavulanic acid presents a stronger selection pressure. A comparison of the absolute fitness values of the wild-type at two temperatures shows that the latter factor has a much larger overall impact on fitness (Supplementary Data), although some variants do show improved absolute fitness at 30 °C (not relative to the wild-type), which is an indication that selection at 30 °C is more favorable for those (Figure S3). However, the rankings of the fittest variants are similar for each library at the two temperatures (Fig. 3), indicating that the fitness landscapes are not very different.

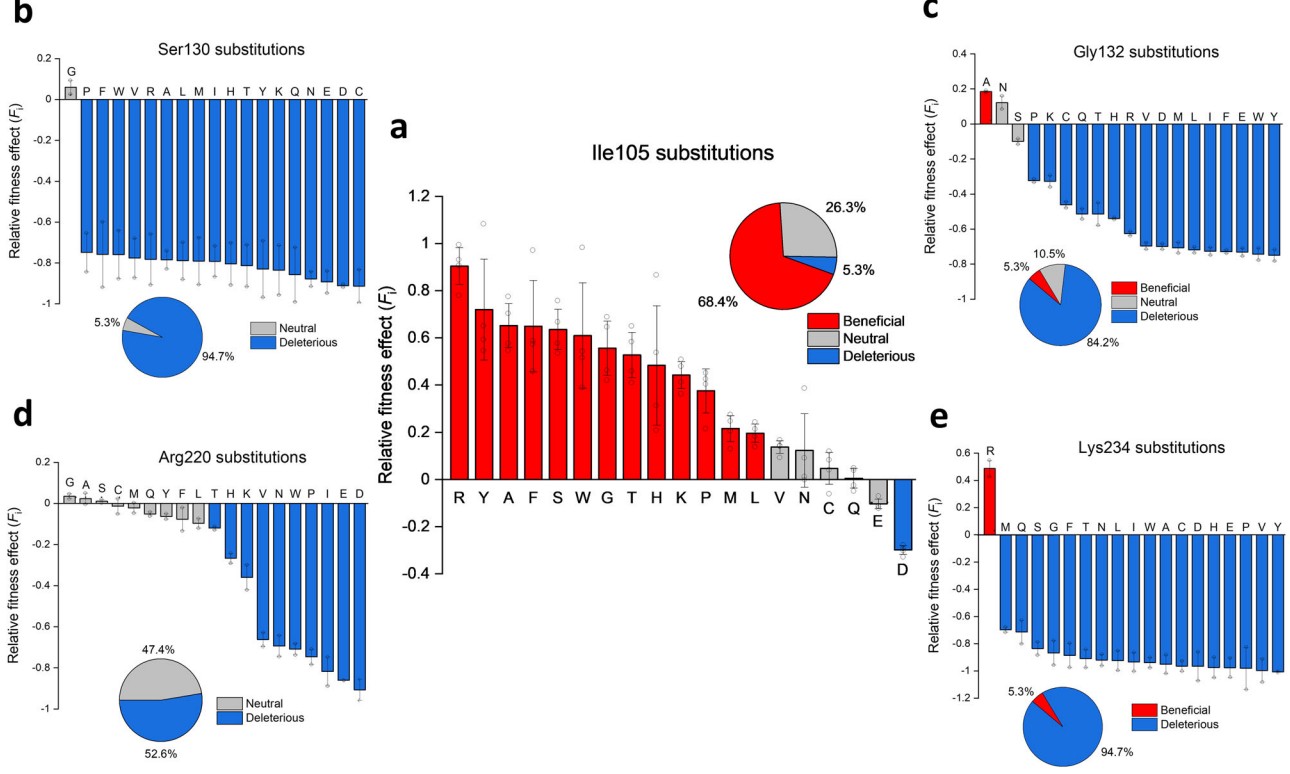

**Fig. 2 | Relative fitness effect ($F_i$) of all single mutant variants selected at 37 °C. a** Ile105 mutants which are present in each library; **b** Ser130 mutants (library 1); **c** Gly132 mutants (library 2); **d** Arg220 mutants (library 3); **e** Lys234 mutants (library 4). Red indicates beneficial variants, blue deleterious and grey variants with a statistically neutral effect on fitness. The degree of reproducibility of the sequencing counts obtained under conditions of no selection in two parallel experiments

provides the basis for defining cutoffs for $F_i$ corresponding to a statistically non-neutral effect on fitness (see Figure S2). Error bars represent SD between four libraries (Ile105 substitutions) or average error between two independent datasets (Ser130, Gly132, Arg220 and Lys234 substitutions). Open circles represent individual data points. Fitness values of all single mutants are available in Supplementary Data.

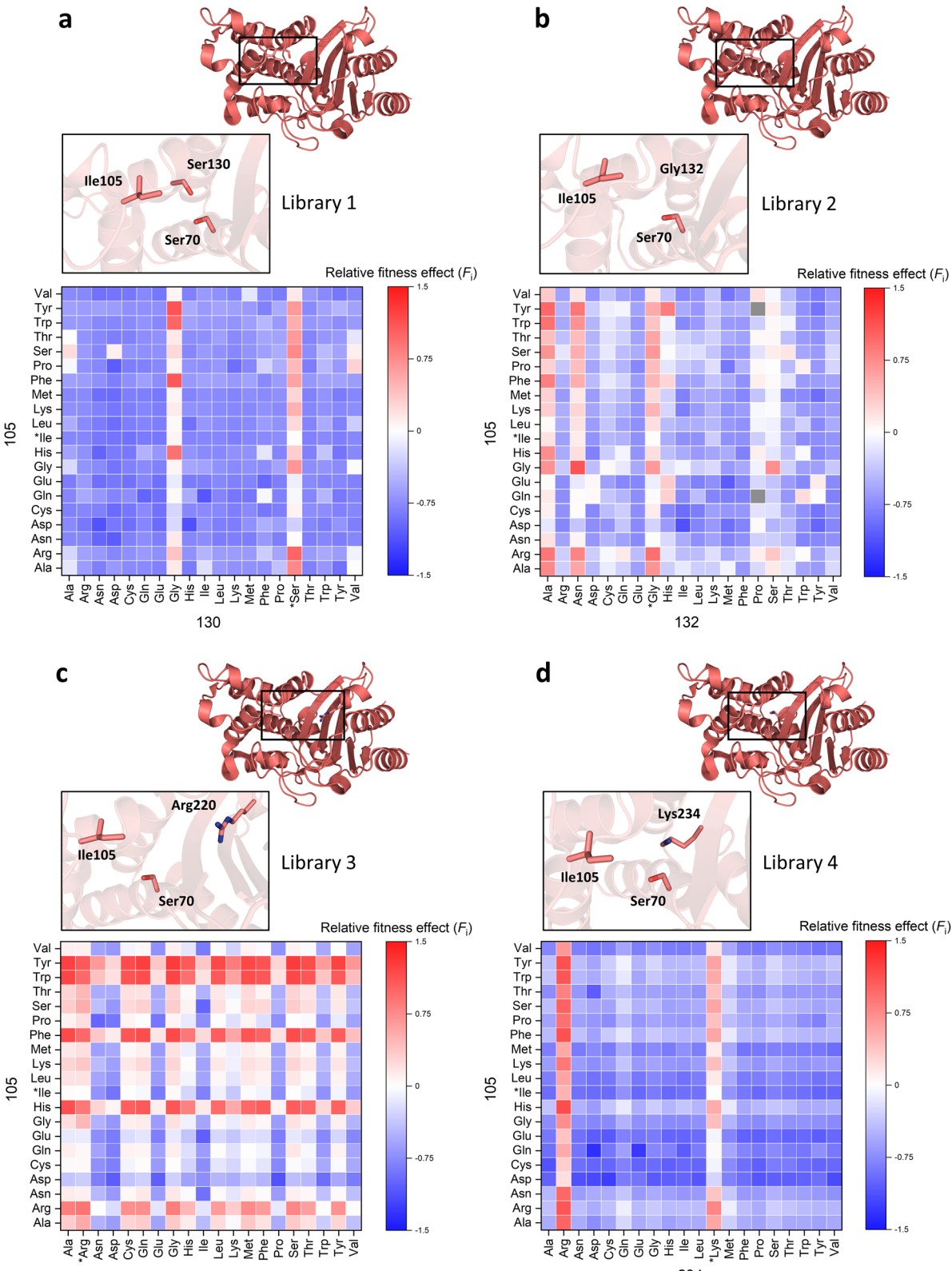

**Fig. 3 | Relative fitness effect of all four library variants selected at 37 °C.**
**a**–**d** Libraries 1-4, positions 105/130 (**a**); 105/132 (**b**); 105/220 (**c**); 105/234 (**d**).
Calculated fitness values for all variants are plotted as heatmaps with red representing positive, blue negative and white neutral fitness effects. Variants for which the fitness effects could not be calculated are colored gray. The vertical axis depicts all possible amino acids on position 105 and the horizontal axis represents all possibilities at positions 130, 132, 220 or 234. Wild-type residues are marked with an asterisk. A closeup view of each mutated residue pair and the catalytic residue Ser70 is shown for each library (PDB 2GDN[16]). A detailed overview of the fitness values can be found in Supplementary Data.

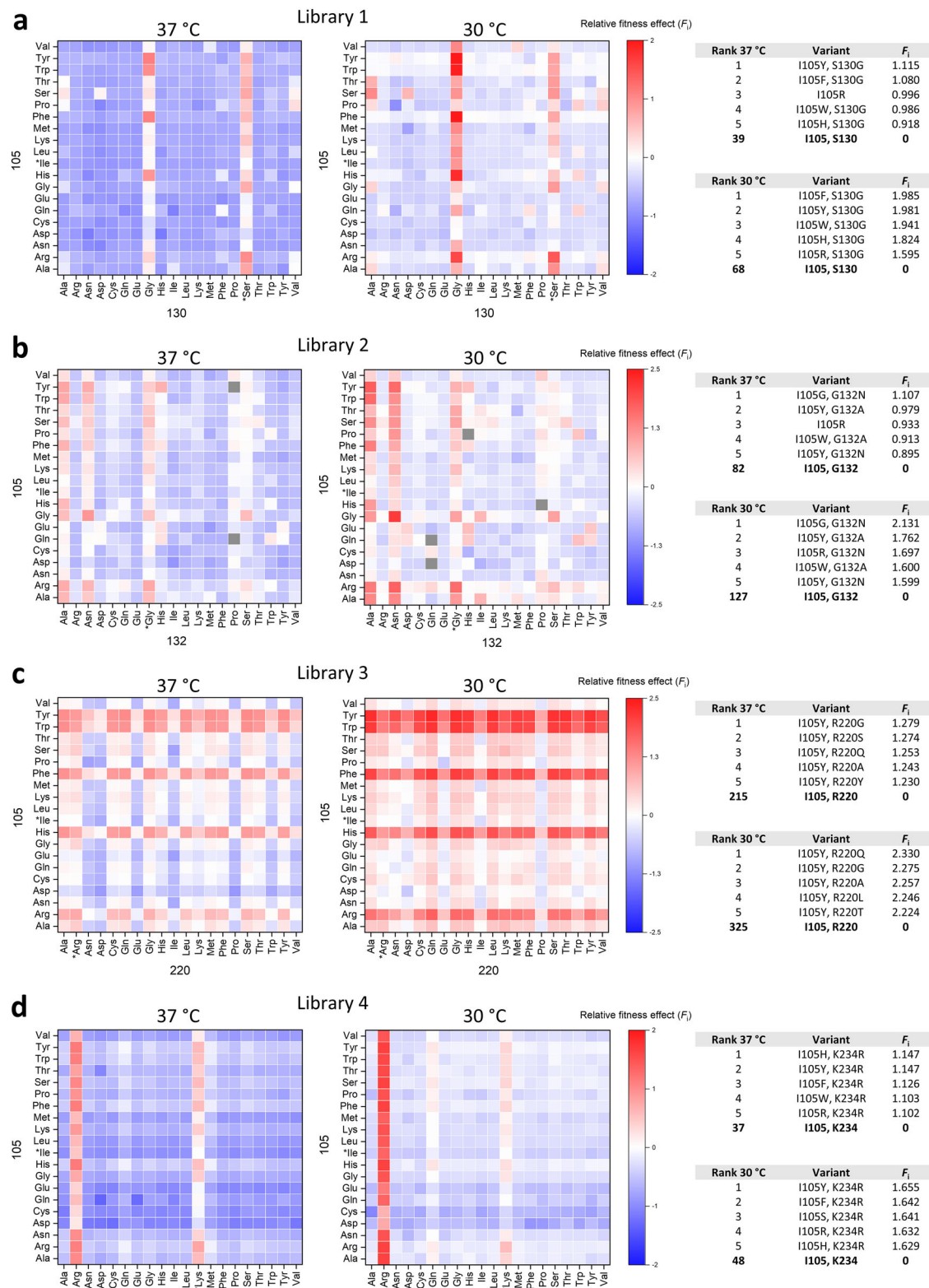

**Fig. 4 | Comparison of relative fitness effects determined at 37 °C and 30 °C. a** Library 1, (**b**) library 2, (**c**) library 3 and (**d**) library 4. Left, heatmaps; right, list of the top five variants together with wild-type and their corresponding $F_i$. Fitness values determined after selection at both temperatures are available in Supplementary Data.

## Pairwise epistasis along evolutionary trajectories

To determine whether the effects of combined single mutations display additive or epistatic phenotypes, we compared fitness values of all double mutants to the sum of values of corresponding single mutants. Pairwise epistasis ($\varepsilon_{AB}$) between mutation A having fitness $W_A$ and mutation B conferring fitness $W_B$ can be defined as shown in Eq. 2:

$$\varepsilon_{AB} = \log \frac{W_{AB}}{W_0} - (\log \frac{W_A}{W_0} + \log \frac{W_B}{W_0}) \qquad (2)$$

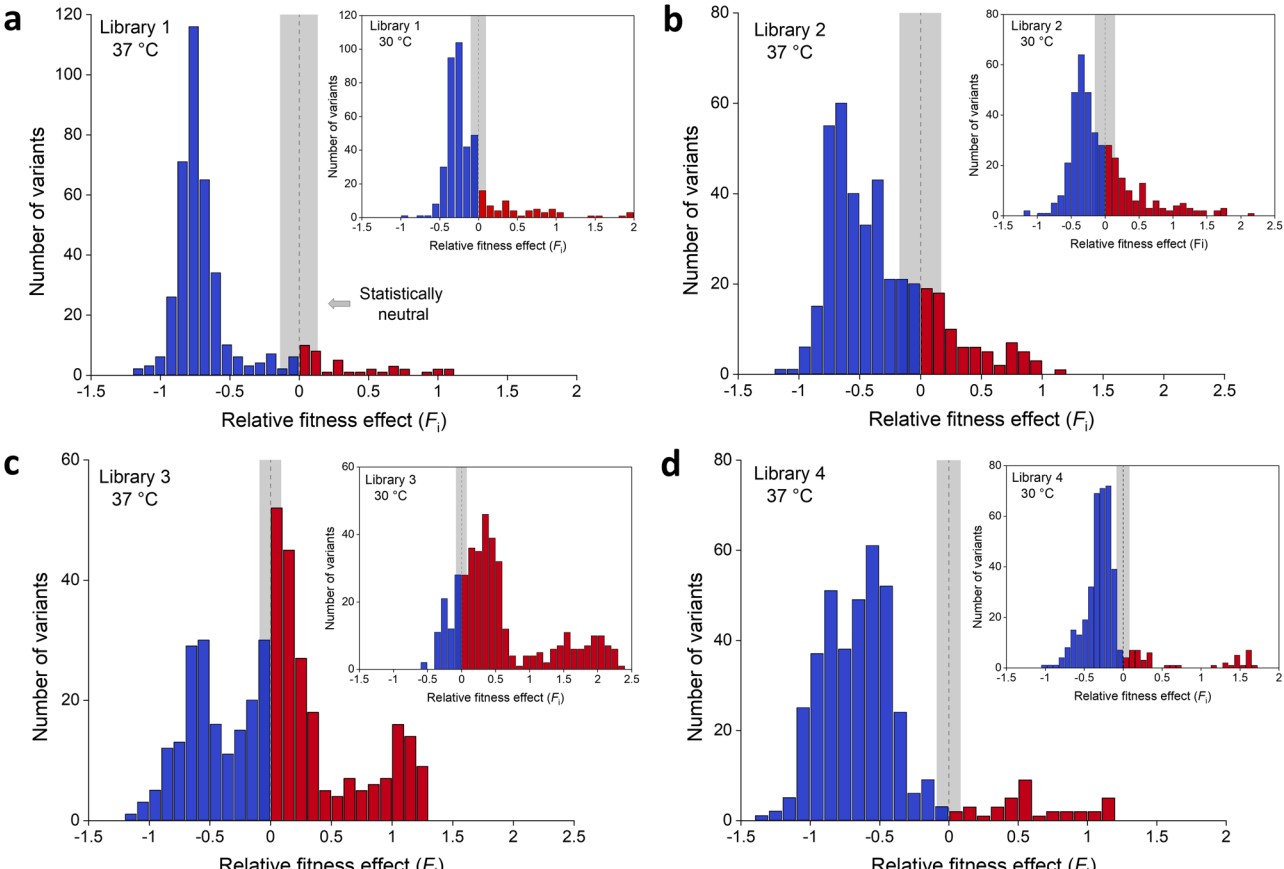

**Fig. 5 | Distribution of fitness effects. a–d** Histograms show that distribution of fitness effects (DFE) is multimodal and library specific. (**a**) Library 1, (**b**) library 2, (**c**) library 3 and (**d**) library 4. Large histograms show DFE for libraries selected at 37 °C; insets show DFE for libraries selected at 30 °C. Variants with negative and positive effects on fitness are colored blue and red, respectively. The dashed line marks zero and the gray highlight corresponds to statistically neutral fitness effects (mean ± two × SD in $F_i^{nt}$; Figure S2).

where $W_0$ and $W_{AB}$ represent wild-type and double-mutant fitness, respectively[37]. The fitness values from Eq. 1 are already logarithmic and relative to the wild-type, so $\varepsilon_{AB}$ can be written as:

$$\varepsilon_{AB} = F_{AB} - [F_A + F_B] \qquad (3)$$

If the relative fitness effects of a double mutant ($F_{AB}$) is higher than that of the sum of the single mutations ($F_A + F_B$), positive pairwise epistasis will be observed and vice versa. Figure 6 shows calculated $\varepsilon_{AB}$ values from all libraries plotted as a function of fitness values of the double mutants at 37 °C and 30 °C. Each library depicts a unique pattern of dispersed epistatic values, with libraries 1,2 and 4 showing a prevailing trend of negative epistasis (Figs. 6a, b and d, respectively). Epistatic values in library 3 (Fig. 6c) are more equally distributed in both directions. Negative epistasis is the most pronounced for variants with negative fitness values, with library 3 being an exception (Fig. 6c). However, it is not rare to observe variants with positive fitness values that exhibit strong negative epistasis, which is most pronounced in libraries 1, 2 and 3 (Figs. 6a, b and c, see the lower right panel of each plot, respectively). Positive epistasis can also occur in variants with negative fitness values, and such variants are found in all four libraries (Fig. 6, upper left panel of each plot). The fittest variants in most cases are a result of positive epistasis, an occurrence which is the most abundant in library 3 (Fig. 6c), and completely absent in library 4 at 37 °C (Fig. 6d). Library 4 has the lowest number of variants displaying positive or negative epistasis, and the average values are also the lowest among all libraries (Fig. 6d). Comparison of total epistasis determined at different temperatures indicates that the average magnitude of negative epistasis increased by 25% at 30 °C, whereas the magnitude of positive epistasis only increased by 5%

(Fig. 7). The total number of variants exhibiting positive or negative epistasis at 30 °C (14.6% and 55.8%) was also elevated compared to 37 °C (11.0% and 48.3%) (Fig. 7).

**Sign epistasis**
Unlike magnitude epistasis, which is calculated from differences between double and single mutants, sign epistasis is determined merely by the sign of fitness values (beneficial or deleterious). Positive sign epistasis can occur for pairs of single mutants containing at least one deleterious mutation with the double mutant being beneficial, and negative sign epistasis occurs when the double mutant displays a detrimental phenotype with at least one of the single mutants having a positive fitness value. Reciprocal sign epistasis is the special case of sign epistasis which occurs when both single mutants are beneficial or deleterious and the double mutant has the opposite sign. We observed a much higher prevalence of negative sign epistasis (53.5%), than positive sign epistasis (4.5%), for libraries screened at 37 °C (Table 1). This observation indicates that on average, deleterious mutations have a much stronger impact on fitness than beneficial ones and are usually found at hotspots for reduced inhibitor sensitivity (Ser130, Gly132, Arg220 and Lys234). Positive sign epistasis was rare in library 1, with only two variants (0.6%), and in library 4, one variant (0.3%), indicating scarcity of evolutionary pathways for the considered residues to climb from fitness valleys formed by deleterious mutations. On the other hand, fitness peaks are most abundant in library 3, where positive sign epistasis is most frequent (15.2%). For libraries screened at 30 °C, the frequency of positive sign epistasis increased by almost a third (6.3% vs. 4.5% at 37 °C), whereas negative sign epistasis was notably less prevalent (53.5% vs. 35.2% at 37 °C). We did not find any cases of reciprocal sign epistasis.

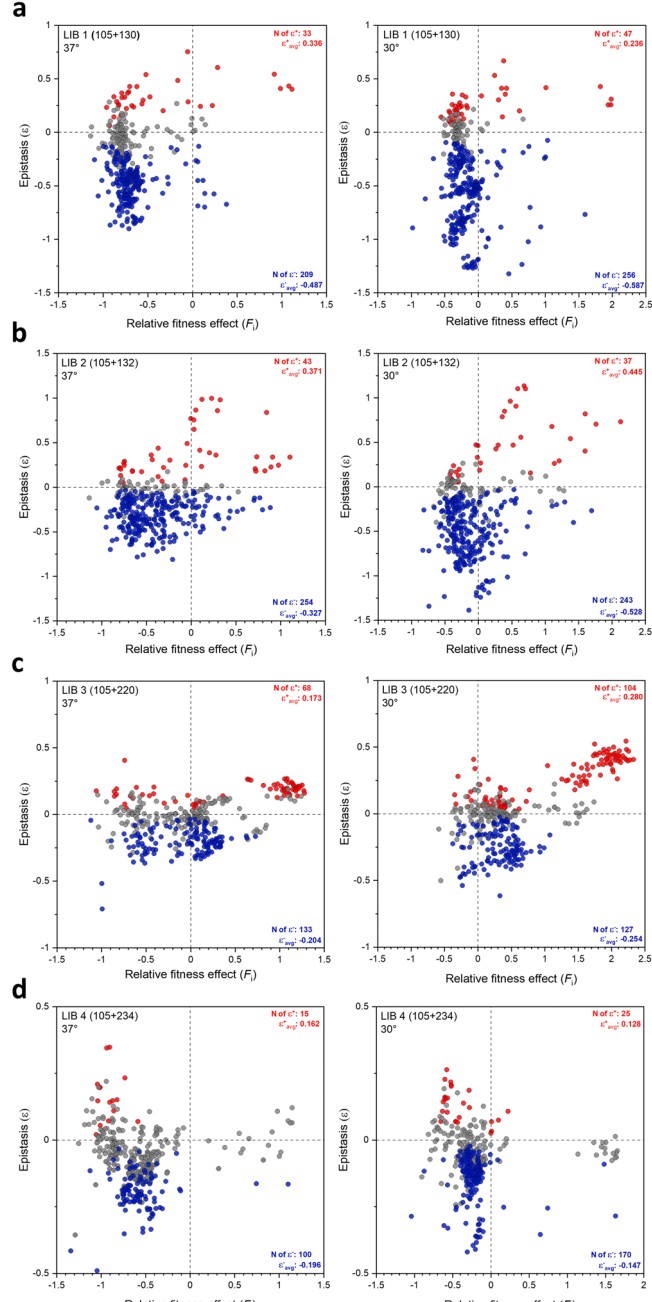

**Fig. 6 | Pairwise epistasis of all double mutant variants. a–d** Epistatic values calculated from Eq. 3 are plotted as a function of fitness values of the double mutants. **a** Library 1, (**b**) library 2, (**c**) library 3 and (**d**) library 4. The $\varepsilon_{AB}$ values are shown for selection at 37 °C (left) and 30 °C (right). Red and blue dots represent significant positive and negative epistatic values, respectively. Non-significant values are shown as gray dots (see Methods for detailed description). All calculated epistatic values are available in Supplementary Data.

## Discussion

β-Lactamases have been researched intensively with a focus on evolutionary studies, using both random and semi-rational approaches to screen for the variants with advantageous traits. Various positions in class A β-lactamases were mutated using saturation mutagenesis[38–41], including 105, 130, 132 and 234[42–46]. Saturating multiple positions simultaneously was also attempted on three consecutive residues in TEM-1 β-lactamase[47,48], and on two positions in SME-1 β-lactamase which are involved in disulfide bond formation[49]. To our knowledge, only one study among class A β-lactamases reported mutagenesis of two positions simultaneously using saturation mutagenesis

and characterization of resulting libraries using deep sequencing[50]. Here, we employed the semi-rational approach in combination with deep sequencing to investigate whether loss of catalytic efficiency caused by inhibitor-resistant mutations can be compensated by another mutation that enhances activity and what role temperature would play in evolutionary process.

Fitness landscapes of all four libraries showed that mutations of the gatekeeper residue Ile105 are capable of complementing mutations at positions Ser130, Gly132, Arg220 and Lys234 which exhibit reduced inhibitor sensitivity, yielding variants with greatly improved in vivo resistance to clavulanic acid inhibition compared to the wild-type (Fig. 3). The number of residues at position 105 that can successfully complement the other mutation is library dependent. However, in most of the cases, aromatic residues (Tyr, Trp, and Phe) as well as His yield the fittest variants (Fig. 3), which is in agreement with literature findings that show enhanced activity of these variants in different β-lactamases[27,44,45]. Unexpectedly, Arg105 is found to be the fittest variant among single mutants (Fig. 2a). We hypothesize that the positive charge of the guanidinium group has favorable electrostatic interactions with the carboxylate group on the C6 substituent of carbenicillin, which was the antibiotic used in the selection process. That would make this residue specific for this substrate, because many other β-lactam compounds do not have a negative charge at the C6 position. Interestingly, combining I105R with the fittest single mutant variants of the other positions results in significant negative epistasis in three out of four libraries (Fig. 8a). The effect is the most pronounced in library 1, where addition of the S130G mutation results in a dramatic reduction of fitness. This implies that, despite its highly beneficial effect as a single mutant, the I105R variant does not pair well with other mutations and might compromise BlaC evolvability. Surprisingly, when averaged over four libraries, 13 variants are significantly better than the wild-type at the gatekeeper position (Fig. 2a). Only amino acids with a negative charge seem to be disfavored (glutamate and aspartate), raising the question why nature selected for Ile at this position in BlaC. In this study, we probed only for activity against carbenicillin and clavulanic acid, so one explanation could be that on average, isoleucine is the best compromise against a myriad other β-lactam antibiotics and inhibitors. Nevertheless, future studies will have to address this notion.

Ser130 is a completely conserved residue among class A β-lactamases[51]. It plays an important role in substrate binding and proton transfer during β-lactam ring opening[43,52], and it is also involved in irreversible crosslinking of mechanism-based inhibitors, including clavulanate[53]. The S130G mutation is known to confer reduced sensitivity to clavulanic acid in several class A enzymes, including BlaC[24,25,43,53]. In our experiment, the S130G variant had a negligible effect on fitness (Fig. 2b). This is in accordance with a previous observation, where S130G did not improve in vivo resistance when introduced in ΔblaC deficient strain of Mtb[24]. Figure 3a shows that the strongest phenotype is observed when Gly is paired with the aromatic residues or His at position 105. High resistance of these variants comes as a result of positive epistasis between these two positions, as fitness of the fittest double mutant is significantly higher than the sum of the individual single variants (Fig. 8b).

Position 132 is also part of the conserved SDN motif, but in BlaC the Asn is substituted for Gly. Restoration of the canonical SDN sequence in BlaC augmented the rate of hydrolysis of clavulanate by four orders of magnitude[26]. BlaC G132S showed significant in vitro and in vivo resistance against sulbactam[54]. Our data shows that Ala is the best option at position 132 and the only variant with a beneficial phenotype in the wild-type background, since mutation to Asn does not result in a significantly higher fitness than the wild-type (Fig. 2c). Surprisingly, when paired with different residues at position 105, the strongest resistance is observed for I105G in combination with G132N; I105G also pairs well with G132S (Fig. 3b). The fittest variant, I105G-G132N, exhibits significant positive epistasis (Fig. 8b). We speculate that the I105G mutation could improve accessibility to the active site, thus allowing BlaC to accommodate substrates more easily. When paired with G132N, the breakdown of both carbenicillin and clavulanic acid could be greatly enhanced.

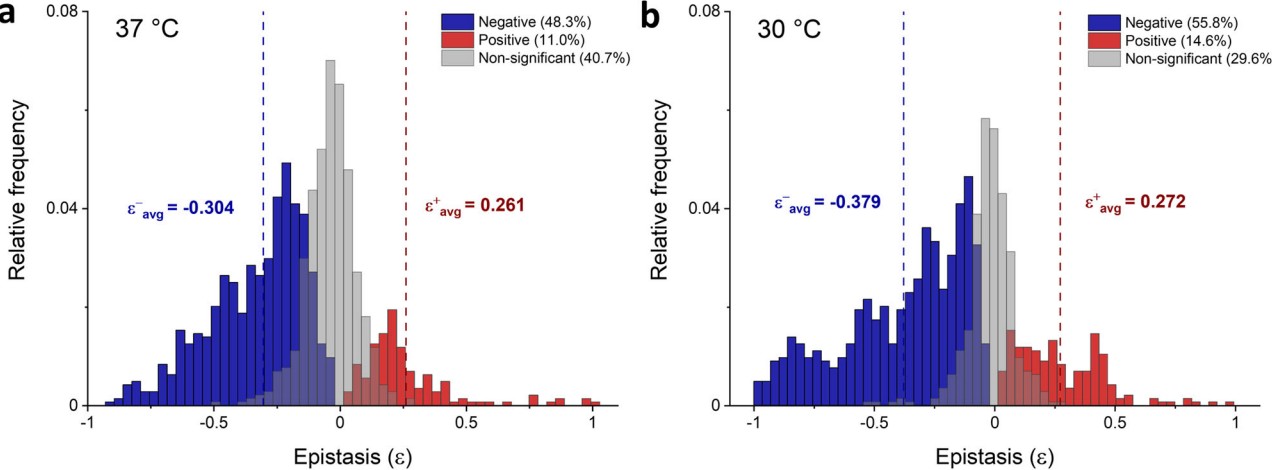

**Fig. 7 | Distribution of epistatic values from all four libraries. a** Libraries selected at 37 °C; (**b**) Libraries selected at 30 °C. Significant epistatic values are indicated in blue and red for negative and positive epistasis, respectively. Dashed lines indicate the average value of negative and positive epistasis. Non-significant epistatic values are colored gray (see Methods for detailed description).

**Table 1 | Summary of sign epistasis for all libraries. Detailed overview of sign epistasis is available in Supplementary Data**

|  | Sign epistasis | | | |
|---|---|---|---|---|
|  | Positive (%) | | Negative (%) | |
|  | 37 °C | 30 °C | 37 °C | 30 °C |
| Library 1 (105 + 130) | 0.6 | 5.8 | 67.3 | 48.8 |
| Library 2 (105 + 132) | 1.9 | 6.9 | 44.9 | 37.7 |
| Library 3 (105 + 220) | 15.2 | 10.8 | 24.1 | 7.5 |
| Library 4 (105 + 234) | 0.3 | 1.7 | 74.5 | 43.5 |
| **Average** | **4.5** | **6.3** | **53.5** | **35.2** |

Arginine at position 220 is not conserved among β-lactamases (45.8%)[51], however, the spatial location of the positive charge is conserved and provided by alternative residues, such as Arg244 in TEM-1. Mutation of Arg220 to either Ser or Ala in BlaC significantly increases resistance against clavulanate, but at the expense of catalytic efficiency[24,25]. This trade-off was reported to be the reason why neither R220S nor R220A improved in vivo resistance when introduced into a Δ*blaC* deficient strain of Mtb, compared to the wild-type[24]. Our data shows that the fitness toll of substituting Arg220 is not as detrimental as in the case of the other three positions (Ser130, Gly132 and Lys234), as many residues can replace Arg220 and still maintain the wild-type level of fitness (Fig. 2d). The fittest variants are again observed when aromatic residues, His or Arg are present at the gatekeeper residue position, generating positive epistatic effects when paired with other mutations. Moreover, aromatic residues and His can even compensate for the negative phenotype caused by amino acids at position 220, as illustrated, for example, by the Asn and Asp columns in Fig. 3c. These observations suggest that mutations of Arg220 alone will not result in highly resistant phenotypes unless complemented with another mutation which ameliorates enzyme activity.

Lys234 is part of the conserved KTG motif in class A β-lactamases, which spans over Ambler positions 234-236 and forms a carboxylate-binding pocket. In TEM-1, mutation K234R reduced affinity for penicillins and clavulanic acid by 5-8 fold, whereas removal of the positive charge at this position in K234T resulted in a catalytically impaired enzyme[55]. When introduced in BlaC, K234R increased resistance against clavulanic acid and, surprisingly, improved catalytic efficiency against penicillin-like β-lactams[25]. Fitness data from library 4 supports previous findings, because besides the wild-type Lys, only Arg emerged as a viable option at position 234 (Fig. 2). Any other amino acid at this position drives fitness towards a deleterious phenotype which cannot be compensated for by any amino acid at position 105 (Fig. 3d). On the other hand, a plethora of options are possible when Lys or Arg are present at 234, and the strongest phenotypes are observed when aromatic residues, His or Arg are present at 105. Contrary to the observations from the other three libraries, the fittest variant I105H-K234R, does not display significant positive epistasis (Fig. 8). Moreover, no variant with a positive effect on fitness is displaying positive epistasis in library 4 after selection at 37 °C (Fig. 6d, left plot) indicating the low epistatic potential of this residue pair.

Among class A β-lactamases, several studies have examined epistasis, both systematically[50,56], and on a small or moderate number of mutants[57,58]. Steinberg et al. determined epistatic landscapes based on an analysis of ~12,500 single amino acid mutants with different "anchor" mutations in the background (E104K or G238S), to simulate an adaptive pathway from TEM-1 to TEM-15, which exhibit cefotaxime resistance[56]. The extent and magnitude of pairwise epistasis strongly depended on the anchor mutation. Only 8% of the mutant pairs with E104K exhibited significant epistasis, compared to 53% with G238S. A follow-up study from the same research group included a comprehensive analysis of 8000 sequential mutant pairs in the TEM-1, showing that negative epistasis (52%) occurred 7.6 times more frequently than positive epistasis (6.8%)[50]. The aim of these studies was to characterize systematically trends of pairwise epistasis throughout an entire protein, without focusing on the epistatic relationship between residues important for function. Schenk et al. looked at the combinations of two sets of four beneficial mutations in TEM-1, which are known to have "large" or "small" effect on cefotaxime resistance[57]. They observed significant negative epistasis in both fitness landscapes, particularly among mutations which had large effects. In the current study, a detailed and complete overview of epistasis is given between both beneficial and deleterious mutations of specific functional residues. We observed the same trend of pervasive negative epistasis (48.3%), which occurred 4.4 times more frequently than positive epistasis (11.0%), for libraries selected at 37 °C. The extent of epistasis seems to be highly library dependent and ranges from 31.9% for library 4, to 82.7% for library 2. Evolution at lower temperature shows that the total number of variants displaying significant epistasis is higher at 30 °C than at 37 °C (70.4% vs 59.3%) (Fig. 7). However, since the extended scope of epistasis is well correlated with generally lower errors of fitness values at 30 °C, which impact significance of epistasis values, we found this difference not to be significant. Values of negative epistasis, however, rose by 25% on average, for libraries selected at 30 °C. The magnitude of negative epistasis primarily arises because of the very poor wild-type fitness, whose contribution on relative fitness of single variants is twice as much compared to the double variants, resulting in "artificially" higher values and thus, more

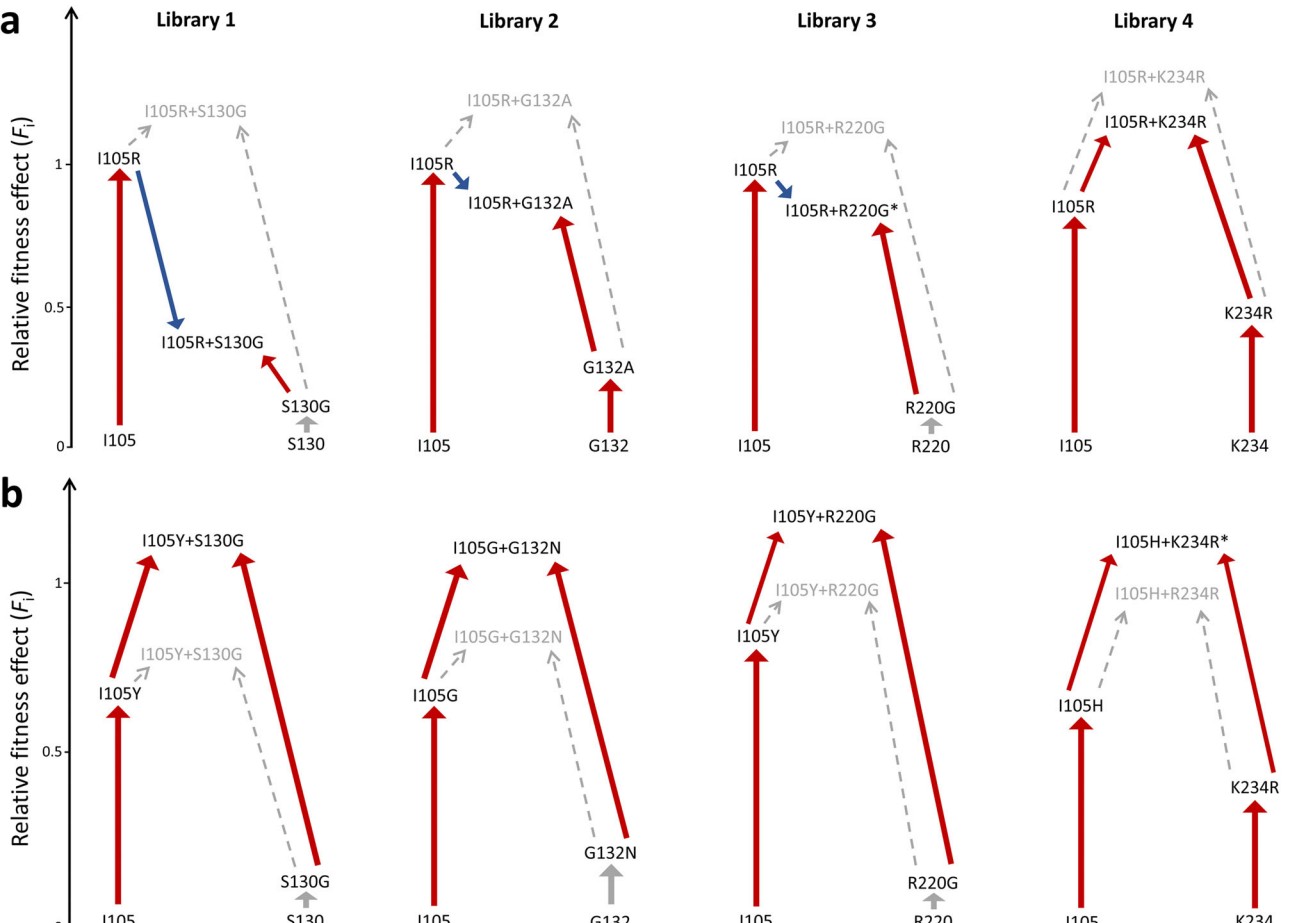

**Fig. 8 | Epistatic effects between position 105 and positions with reduced inhibitor sensitivity observed after selection at 37 °C. a** Negative epistasis is observed in three cases for combinations of the I105R mutation with the fittest single mutant variants of positions with reduced inhibitor sensitivity; **b** Positive epistasis observed for the fittest variants of the four libraries. Red, blue, and gray arrows represent positive, negative and neutral shifts in fitness, respectively. Dashed arrows indicate predicted fitness from the sum of single mutant fitnesses. Asterisks denote variants with non-significant epistasis.

negative epistasis (see Eq. 1 and 3). The same phenomenon is accountable for the increased frequency of positive sign epistasis and diminished incidences of negative sign epistasis at lower temperature (Table 1). The frequency of positive sign epistasis within each library is well correlated with the mutational tolerance of positions with reduced inhibitor sensitivity, indicating that mutations of more conserved residues are more detrimental and cannot be easily compensated for by another mutation. An equal trend of fitness compensation between libraries was observed when we expanded our criteria to all double mutants with either beneficial or neutral effect on fitness, with one of the single mutants being deleterious (Table S2).

In conclusion, with the exception of G132A and K234R variants, we observed that mutations of positions associated with reduced inhibitor sensitivity do not significantly improve fitness at a relatively low selection pressure of clavulanic acid (MIC of the wild-type). On the other hand, mutations of the gatekeeper position have a highly beneficial effect on fitness and interact synergistically with mutations of other positions, which produces the fittest phenotypes, often as a result of positive epistasis. Mutational landscapes generated at 37 °C and 30 °C were similar, indicating that in this experiment temperature did not significantly influence evolutionary outcomes against clavulanic acid and carbenicillin. Our results underline the possibly important role of epistasis in the evolution of functional residues and provide a solid foundation for future studies with the aim of elucidating more complex networks of interactions between different residues in β-lactamases and their potential implications in antibiotic resistance.

## Methods

### Multi-site saturation mutagenesis

Four double-site saturation mutagenesis libraries were constructed by utilizing the nicking mutagenesis method[32]. Mutagenic oligonucleotides containing either NNK or NNS degenerate codons were targeted to five positions in the *blaC* gene (I105, S130, G132, R220 and K234, Table S1). The original protocol for multi-site saturation mutagenesis was modified according to Kirby et al[59]. The time of mutagenesis strand synthesis was extended from 20 min to 60 min. A detailed description of the library preparations is provided in the Supplementary note 1.

### Clavulanic acid selection

All selection experiments were performed in *E. coli* strain KA797[60]. *E. coli* cells were transformed by electroporation with 100 ng of each plasmid library, plated on nonselective media (LB-agar with 50 µg/mL kanamycin) and incubated overnight at 37 °C. Each transformation yielded more than $10^7$ transformants. Cells were recovered from agar plates with 10 mL of LB broth and diluted 1:20, after which $OD_{600}$ was measured in triplicate. These cultures were used to inoculate 20 mL of nonselective media (LB broth with 50 µg/mL of kanamycin) and 20 ml of selective media (LB broth with 50 µg/mL of kanamycin, 100 µg/mL of carbenicillin and 1 µg/mL of clavulanic acid) to have an $OD_{600}$ = 0.01. The cultures were incubated for 2.5 h at 37 °C or 4 h at 30 °C, and then put on ice for 15 min to stop the growth. The incubation time was chosen to obtain significant selection, while maintaining a sufficient population size relative to the library diversity and

avoiding the stationary phase. The cultures were centrifuged, washed twice by resuspension in 20 mL LB broth, then centrifuged again and resuspended in 3 mL LB broth with 50 μg/mL of kanamycin and incubated overnight to saturation at 37 °C. Next morning, plasmid libraries were harvested using GeneJET Plasmid Miniprep Kit (Thermo Fischer Scientific). Two independent experiments were performed, starting with the transformation of *E. coli* with plasmid libraries.

## Deep sequencing

Samples for Illumina sequencing were prepared by PCR using both selected and unselected libraries as templates. Amplicons were barcoded using universal tail sequences and sequenced by a commercial partner on either an Illumina MiSeq or NovaSeq 6000 sequencer (libraries 3 and 4, and libraries 1 and 2, respectively). MiSeq samples were sequenced paired-end 300-bp using 600 cycle v3 sequencing reagents. NovaSeq samples were sequenced paired-end 150-bp using 300 cycle v1.5 sequencing reagents. FASTQ files from both systems were generated using bcl2fastq v2.20.

## Data analysis

The sequencing data were analyzed using the Galaxy open web-based platform[61–63]. Paired-end 300-bp MiSeq reads were merged with a minimum overlap of 50 bp and maximum mismatch density of 0.25. Reads were filtered for quality score (reads with >10% of base calls with quality score less than 20, or probability of error = $10^{-2}$, were discarded) and length (only accepting reads with 465 nt). Identical reads were collapsed into a single read in FASTA format and then trimmed to only 6 nt (codons for position 105, and 220 or 234) and the frequency of each variant under each condition and temperature was determined. Paired-end 150-bp NovaSeq reads were analyzed in the same manner, but without merging, with a different length filter (only accepting reads with 150 nt) and different trimmed positions (codons for position 105, and 130 or 132).

Epistasis values were determined to be significantly positive or significantly negative based on Eqs. (4) and (5), respectively:

$$\varepsilon_{AB} = F_{AB} - (F_A + F_B + \Delta F_A + \Delta F_B + \Delta F_{AB}) > 0 \qquad (4)$$

$$\varepsilon_{AB} = F_{AB} - (F_A + F_B + \Delta F_A + \Delta F_B + \Delta F_{AB}) < 0 \qquad (5)$$

where $F_{AB}$, $F_A$ and $F_B$ stand for fitness of the double mutant and corresponding single mutants, and $\Delta F_{AB}$, $\Delta F_A$ and $\Delta F_B$ represent their corresponding errors between two replicate datasets.

Sign epistasis was determined solely based on fitness values of the individual mutations and their double mutant pair. Positive sign epistasis emerged when at least one of the mutants was individually deleterious (below the lower limit of statistically neutral cutoff), and the double mutant was beneficial (above the upper limit of statistically neutral cutoff). Negative sign epistasis occurred when at least one of the mutants was individually beneficial, and the double mutant was deleterious. Reciprocal sign epistasis demands both mutants to be individually deleterious, while the double mutant is beneficial. Negative reciprocal sign epistasis is the opposite.

## Reporting summary

Further information on research design is available in the Nature Portfolio Reporting Summary linked to this article.

## Data availability

Raw sequencing files are available through the NCBI SRA, under BioProject accession PRJNA1048706. Sequencing counts following quality filtering and calculated fitness and epistatic values are available in Supplementary Data.

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

## Acknowledgements
We thank Dr Aimee L. Boyle for critical reading of the manuscript and Leiden Genome Technology Center for the assistance with deep sequencing. This project was supported by the Dutch Research Council, project OCENW.KLEIN.264 (to M.U.).

## Author contributions
M.R performed experiments, analyzed data, wrote, and edited the manuscript. M.U. obtained funding, supervised experiments, analyzed data, and edited the manuscript.

## Competing interests
The authors declare no competing interests.
