## [Peer Review File · Communications Biology]

Reviewers' comments:

Reviewer #1 (Remarks to the Author):

This manuscript by Radojković and Ubbink delves into the extent to which the loss of catalytic activity, induced by mutations that reduce inhibition by clavulanic acid (130, 132, 220, 234), can be compensated by an activity-conferring mutation (105). The authors accomplish this by screening four double site-saturation libraries (105-130, 105-132, 105-220, 105-234), both in the absence and presence of the drug combination carbenicillin and clavulanic acid, evaluating the relative fitness of all mutants (#1540). Their findings reveal that most 105 mutations increase fitness, while substitutions at 130, 132, 220, and 234 mostly result in neutral or deleterious effects. Additionally, the double mutant landscape illustrates that certain 105 substitutions can compensate for the fitness loss induced by mutations at 130, 132, 220, and 234. The manuscript concludes with an exploration of the epistatic interactions between these mutations, highlighting pervasive negative epistasis.

Overall, the manuscript provides valuable insights into the adaptive responses of BlaC β -lactamase to clavulanic acid. The study's topic holds general interest for molecular evolutionary biologists. The results are effectively presented. The methods are clear, and the statistical analyses are appropriate. While the experimental observations are not entirely surprising, they describe a novel system. However, the paper reads somewhat descriptive, and the extent of the expanded discussion on the role of each focal mutation could be reconsidered.

Comments:

My primary concern with the study lies in the fact that the authors did not directly assess activity loss at positions 130, 132, 220, and 234 and their subsequent compensation. Rather, the focus was on fitness effects and how mutations at amino acid position 105 could compensate for them. The manuscript would benefit from a discussion on whether fitness can be reliably used as a proxy for enzyme activity. Given that in Kurz et al (PMID: 24060876), R220A, R220S, and S130G exhibit reduced catalytic efficiency, despite seemingly unaffected fitness in this study, raises questions about the relationship between fitness and enzyme activity. Addressing the validity and limitations of using fitness as an indicator of enzyme activity would enhance the interpretation of the results.

The title implies a process of evolutionary change over time. However, it's worth noting that the study predominantly involves screening a library of in vitro generated mutants rather than observing natural evolution. While the term "evolution" can be interpreted broadly, the nature of the experimental design and results may be more accurately captured by an alternative title.

Additionally, the title mentions the compensation of activity loss, but considering the absence of direct kinetic measurements and reliance on fitness assessments, I find the assertion of activity loss compensation less substantiated.

Additionally, it would be valuable to explicitly discuss the extent of compensation for fitness loss. While some information can be inferred from Figures 3A-D, providing a more explicit measure, such as the frequency of mutants that are compensated, would enhance the understanding of the prevalence of compensatory mutations.

Minor comments/ suggestions:

L137, L177: The authors refer to the libraries as evolved, consider using "selected" for a more precise characterization.

Reviewer #2 (Remarks to the Author):

Radojkovic and Ubbinks analyzed the effect of compensatory mutations through four mutant libraries of BlaC β -lactamase, combining the variants of I105 with the ones of Ser130, Gly132, Arg220 or Lys234. They evaluated the effect of the single variants on fitness, as well as the effect of the double variants, and analyzed the mutational landscape at 37°C and 30°C. This study is relevant since it explores the role of epistasis in the evolution of BlaC β -lactamase, and because of its potential implication in antibiotic resistance. In addition, the study is original, well written and very well presented, I wish I always received papers of such caliber. Nonetheless, I have some comments to improve the overall quality of the manuscript.

-According to material and methods, the transformants were incubated 2.5 h at 37 °C and 4 h at 30°C. This step, as you explained, is a selection or a fitness assay, but I would not consider it as an evolution experiment (Line 136). In several sentences of the manuscript, you refer that the libraries "evolved at 37°C or 30°C, e.g line 136, 167, 177. From my point of view, the world "evolved" should be replaced by "obtained", "selected" or similar. The same situation happens with the title "Evolution of clavulanic acid resistance ..."; the mutants were created by saturating the selected positions, there was not an evolutionary step.

- Have you evaluated the kinetic parameters and stability of the fittest double mutants of the four libraries?

- Has any of the fittest double mutants been found in clinical isolates?

For the study, you have used *E.coli*, although BlaC (CG%: 66%) is typically found in *Mycobacterium tuberculosis*. *M. tuberculosis* presents a GC% content higher than the one of *E. coli* (65.6% vs 50.8%). For this, I am wondering if the results obtained in *E. coli* could be representative of the ones that you could obtain in *M. tuberculosis*, or a bacterium closer to *M. tuberculosis* could have been a better option for representing what happens in the natural strain.

-Line 80-81: Add a reference to support the statement.

-Lines 222-236: replace "Figure 5" by "Figure 6" in the whole paragraph.

-Line 326: replace "Figure 3B" by "Figure 3A".

-Line 370 replace "Figure 3C" by "Figure 3D".

Reviewer 1

1. *My primary concern with the study lies in the fact that the authors did not directly assess activity loss at positions 130, 132, 220, and 234 and their subsequent compensation. Rather, the focus was on fitness effects and how mutations at amino acid position 105 could compensate for them. The manuscript would benefit from a discussion on whether fitness can be reliably used as a proxy for enzyme activity. Given that in Kurz et al (PMID: 24060876), R220A, R220S, and S130G exhibit reduced catalytic efficiency, despite seemingly unaffected fitness in this study, raises questions about the relationship between fitness and enzyme activity. Addressing the validity and limitations of using fitness as an indicator of enzyme activity would enhance the interpretation of the results.*

In our study, bacterial fitness is dependent upon two factors: BlaC ability to effectively hydrolyse β -lactam antibiotic and ability to evade inhibition by β -lactam inhibitor (this definition has been added at the beginning of the Results section, L104-106). Thus, the resulting fitness is a consequence of the combination of these two traits. Variants described in Kurz et al. (PMID: 24060876) (R220A, R220S and S130G) exhibit reduced inhibitor-sensitivity, but also reduced catalytic efficiency. In line with these findings we find that the fitness of these variants is not significantly different from the wild-type (Figure 2).

Furthermore, mutation of gatekeeper residue I105F has been shown to enhance catalytic efficiency of BlaC by 3-fold (ref. ⁵). Some of the fittest double mutant variants in our study also contain this mutation, along with I105Y, I105W and I105H, which are variants with highest penicillinase activity in other β -lactamases (ref ⁶⁻⁸). The effect of these mutations is primarily to increase activity although a small part may be due a higher level of the protein caused by a slight stabilization effect.

Using bacterial fitness as a proxy for β -lactamase activity has been employed throughout literature (ref. ¹⁻³, also added in the main text). It has been shown that evolution of a new function is primarily driven by improvement in enzyme efficiency toward specific substrate, rather than increased thermostability (ref. ⁴).

2. *The title implies a process of evolutionary change over time. However, it's worth noting that the study predominantly involves screening a library of in vitro generated mutants rather than observing natural evolution. While the term "evolution" can be interpreted broadly, the nature of the experimental design and results may be more accurately captured by an alternative title. Additionally, the title mentions the compensation of activity loss, but considering the absence of direct kinetic measurements and reliance on fitness assessments, I find the assertion of activity loss compensation less substantiated.*

The title has been changed to “Positive epistasis drives clavulanic acid resistance in double mutant libraries of BlaC β -lactamase”.

3. *Additionally, it would be valuable to explicitly discuss the extent of compensation for fitness loss. While some information can be inferred from Figures 3A-D, providing a more explicit measure, such as the frequency of mutants that are compensated, would enhance the understanding of the prevalence of compensatory mutations.*

Compensation of fitness loss is demonstrated by positive sign epistasis in the last part of Results section (L256-L275) and the observed frequency for each library is given in the Table 1. The most noteworthy observations are then further discussed in the Discussion section (Role of Arg220, Role of Lys234 and Pairwise epistasis). Additionally, we provided frequency of all compensated mutants with either beneficial or neutral effect on fitness in Table S2, which is discussed in main text at L419-L422.

4. *L137, L177: The authors refer to the libraries as evolved, consider using "selected" for a more precise characterization.*

Suggested change has been done at L137, L177 and elsewhere throughout the manuscript.

Reviewer 2

1. *According to material and methods, the transformants were incubated 2.5 h at 37 ° C and 4 h at 30 ° C. This step, as you explained, is a selection or a fitness assay, but I would not consider it as an evolution experiment (Line 136). In several sentences of the manuscript, you refer that the libraries “evolved at 37 ° C or 30 ° C, e.g line 136, 167, 177. From my point of view, the word “evolved” should be replaced by “obtained”, “selected” or similar. The same situation happens with the title “Evolution of clavulanic acid resistance ...”; the mutants were created by saturating the selected positions, there was not an evolutionary step.*

As suggested, verb “evolved” has been replaced with more suitable verbs “selected” or “obtained” at lines 136, 167, 177 and elsewhere throughout the manuscript. New title is “Positive epistasis drives clavulanic acid resistance in double mutant libraries of BlaC β -lactamase”.

2. *Have you evaluated the kinetic parameters and stability of the fittest double mutants of the four libraries?*

We have done extensive *in vitro* characterization of some the fittest double mutants and those results are planned for another manuscript. We can report already that I105Y+S130G and I105G+G132N exhibit positive epistatic compensation of activity loss when comparing k_{cat}/K_m values to those of single mutants. Additionally, I105G+G132N displays significant increase in thermostability. Both double mutants

also retain the reduced-inhibitor sensitivity profile of their single mutant counterparts (S130G and G132N), although to a lesser extent.

3. *Has any of the fittest double mutants been found in clinical isolates?*

To our knowledge, none of the fittest double mutants has been reported in the literature. Contrary to the expectations, observed polymorphism of *blaC* gene in clinical isolates is usually associated with increased susceptibility to β -lactams (example mutations, ambler numbering: A49G, S99R, L138V, and G156S; ref. ⁹⁻¹¹). This can be explained by still infrequent use of β -lactam antibiotic/inhibitor combinations to treat tuberculosis and thus, absence of high selection pressure. Nonetheless, if demand for their use rises in the future, we would not be surprised if one of the fittest double mutant variants described in this study occurs in clinical isolates.

4. *For the study, you have used E. coli, although BlaC (CG%: 66%) is typically found in Mycobacterium tuberculosis. M. tuberculosis presents a GC% content higher than the one of E. coli (65.6% vs 50.8%). For this, I am wondering if the results obtained in E. coli could be representative of the ones that you could obtain in M. tuberculosis, or a bacterium closer to M. tuberculosis could have been a better option for representing what happens in the natural strain.*

We do not see how the GC content has bearings on the results because the fitness is based on the phenotype that is based on the activity of the β -lactamase.

In our lab, we observed good correlation between resistance profiles of several BlaC variants produced in *E. coli* and *Mycobacterium marinum* (ref. ¹²). Additionally, we use construct with twin-arginine translocation (TAT) system for exporting BlaC to the periplasm of *E. coli*, which is the same export pathway in Mtb. Hence, we have solid reason to believe that results obtained for *E. coli* are representative of what would happen in bacterium close to Mtb. Use of *M. marinum* in this study would not be appropriate, because we wanted to compare fitness landscapes generated at two temperatures (30 °C and 37 °C) and optimal temperature for *M. marinum* growth is between 30-33 °C.

5. *Line 80-81: Add a reference to support the statement.*

The statement is now supported by the references.

6. *Lines 222-236: replace “Figure 5 ” by “Figure 6 ” in the whole paragraph.*

Figure 5 was replaced by Figure 6 in the whole paragraph.

7. *Line 326: replace “Figure 3B ” by “Figure 3A ”.*

Figure 3B was replaced by Figure 3A.

8. Line 370 replace “Figure 3C” by “Figure 3D”.

Figure 3C was replaced by Figure 3D.

References

1. Bershtein, S., Segal, M., Bekerman, R., Tokuriki, N. & Tawfik, D. S. Robustness-epistasis link shapes the fitness landscape of a randomly drifting protein. *Nature* **444**, 929–932 (2006).
2. Firnberg, E., Labonte, J. W., Gray, J. J. & Ostermeier, M. A comprehensive, high-resolution map of a Gene’s fitness landscape. *Mol. Biol. Evol.* **31**, 1581–1592 (2014).
3. Stiffler, M. A., Hekstra, D. R. & Ranganathan, R. Evolvability as a function of purifying selection in TEM-1 β -lactamase. *Cell* **160**, 882–892 (2015).
4. Knies, J. L., Cai, F. & Weinreich, D. M. Enzyme efficiency but not thermostability drives cefotaxime resistance evolution in TEM-1 β -lactamase. *Mol. Biol. Evol.* **34**, 1040–1054 (2017).
5. Feiler, C. *et al.* Directed evolution of Mycobacterium tuberculosis β -lactamase reveals gatekeeper residue that regulates antibiotic resistance and catalytic efficiency. *PLoS One* **8**, (2013).
6. Doucet, N., De Wals, P. Y. & Pelletier, J. N. Site-saturation mutagenesis of Tyr-105 reveals its importance in substrate stabilization and discrimination in TEM-1 β -lactamase. *J. Biol. Chem.* **279**, 46295–46303 (2004).
7. Papp-Wallace, K. M. *et al.* Elucidating the role of Trp105 in the KPC-2 β -lactamase. *Protein Sci.* **19**, 1714–1727 (2010).
8. Judge, A. *et al.* Mapping the determinants of catalysis and substrate specificity of the antibiotic resistance enzyme CTX-M β -lactamase. *Commun. Biol.* **2023 61** **6**, 1–11 (2023).
9. Olivença, F. *et al.* Uncovering beta-lactam susceptibility patterns in clinical isolates of Mycobacterium tuberculosis through whole-genome sequencing. *Microbiol. Spectr.* **10**, (2022).
10. Li, F. *et al.* In vitro activity of β -lactams in combination with β -lactamase inhibitors against Mycobacterium tuberculosis clinical isolates. *Biomed Res. Int.* **2018**, (2018).
11. Park, S., Jung, J., Kim, J., Han, S. B. & Ryoo, S. Susceptibility of β -Lactam Antibiotics and Genetic Mutation of Drug-Resistant *Mycobacterium tuberculosis* Isolates in Korea. *Tuberc. Respir. Dis. (Seoul)*. **85**, 256–263 (2022).
12. van Alen, I. *et al.* Mycobacterium tuberculosis β -lactamase variant reduces sensitivity to ampicillin/avibactam in a zebrafish-*Mycobacterium marinum* model of tuberculosis. *Sci. Reports 2023 131* **13**, 1–9 (2023).

REVIEWERS' COMMENTS:

Reviewer #1 (Remarks to the Author):

Thank you for the detailed response to my comments. I have thoroughly reviewed the rebuttal, and I appreciate the clarifications and changes made to address my concerns. I am satisfied with the modifications implemented, and I believe the manuscript is now more robust.

Considering the improvements and adjustments made in response to the comments, I suggest the acceptance for submission of the manuscript.

Reviewer #2 (Remarks to the author):

The quality of the manuscript has improved following the reviewers suggestions. In my opinion, the manuscript can be accepted in this version.

Sincerely

Laura Dabos